# Mpox Patient Journey in Israel

**DOI:** 10.3390/microorganisms11041042

**Published:** 2023-04-16

**Authors:** Tal Patalon, Galit Perez, Yaki Saciuk, Ziva Refaeli, Sivan Gazit

**Affiliations:** 1Kahn Sagol Maccabi (KSM) Research & Innovation Center, Maccabi Healthcare Services, Tel Aviv 68125, Israel; 2Maccabitech Institute for Research and Innovation, Maccabi Healthcare Services, Tel Aviv 68125, Israel; 3Department of Health Policy and Management, School of Public Health, Ben Gurion University of the Negev, Beer Sheva 84105, Israel

**Keywords:** Mpox, MPX, monkeypox, patient journey, Orthopoxvirus genus

## Abstract

Reports on Mpox have, thus far, characterized the disease, but mostly through a single timepoint view. The aim of this study was to characterize Mpox in the Israeli setting, in general, alongside compiling a detailed patient journey from multiple in-depth interviews with infected individuals. This descriptive study followed two complimentary paths: retrospective and prospective. The first consisted of interviews with Mpox patients, while the retrospective part included the retrieval of anonymized electronic medical records of patients diagnosed with Mpox between May and November 2022. Patient characteristics in Israel were, overall, comparable to global reports. We found that the median time from symptoms to first suspicion of Mpox was 3.5 days, while the median time from the first symptom to a confirmatory test was 6.5 days, which could explain the surge in Israel. The duration of lesions did not alter in terms of their anatomical location, while lower Ct values correlated both with a longer symptom duration and more symptoms. Most patients reported anxiety to a high degree. Clinical trials that consist of a long-term relationship with the medical researchers contribute greatly to a deeper understanding of the patient journey, especially for unfamiliar or stigmatized diseases. Emerging infections, such as Mpox, should be further investigated to assess asymptomatic carriers, especially when rapidly spreading.

## 1. Introduction

The recent surge in Mpox (formerly referred to as monkeypox) disease is characterized by a wider geographic distribution than the one seen between the 1970s, when it was first identified in humans, and 2022 (from Africa, mostly the Democratic Republic of Congo, to Europe and the United states) [1]. At the time of writing, since May 2022, 86,231 cases have been reported, more than 98% of them in locations that have not historically reported Mpox [2]. In Israel, 262 cases have been confirmed as of 31 October 2022 [3], after which the spread seems to have largely stopped.

Apart from the geography, an alteration in the sociodemographic characteristics of patients with Mpox has occurred as well, from children to mostly young men who identify as gay, bisexual, and other men who have sex with men (GBMSM), whereas the mode of transmission (previously including animal-to-human transmission) is now probably primarily propagated through human-to-human transmission [4]. Previous reports have also demonstrated a different clinical course of the current outbreak (also referred to as atypical presentation), such as asynchronous lesion development and localization of lesions to the genital, perineal/perianal, or perioral areas [5].

Reports on Mpox have, thus far, characterized the disease, but mostly through a single timepoint view, rather than a patient journey, except for case reports. To this point, the aim of this study is to characterize Mpox in the Israeli setting, in general, alongside compiling a detailed patient journey from multiple in-depth interviews with infected individuals.

## 2. Materials and Methods

The study followed two complimentary paths: a retrospective one and a prospective one, both conducted in the Maccabi Healthcare Service (MHS) research center. MHS is the second-largest nationwide, not-for-profit health fund in Israel, insuring and providing medical services to over 26% of the population. Membership in one of the four national health funds is mandatory; all individuals freely choose a health fund, each of which is obliged to offer membership to any Israeli.

The prospective part comprised a clinical trial, which included interviews between a certified nurse and MHS members infected with Mpox. Once a patient received a diagnosis of Mpox, their MHS family physician was contacted by a researcher from the MHS Research Center to explain the study and obtain their consent to contact the patient by phone. Approaching the primary care practitioner prior to contacting the patient directly is a policy dictated by the MHS’ Institutional Review Board (ethics committee) for clinical trials at MHS. Once such type of consent was obtained, a nurse from the MHS Research Center contacted the Mpox patient by phone, explaining the study and obtaining informed consent. Then, consenting members were asked about the symptoms they experienced as well as their journey to diagnosis. The nurse followed-up with a phone conversation every 1 to 4 days for the next two weeks, after which a weekly to monthly call was performed, depending on the patients’ course of disease and preference. The reported symptoms were collected for each day, in order to be later aggregated for descriptive analysis.

The retrospective part consisted of the retrieval of anonymized electronic medical records (EMRs) from the MHS centralized computerized database of any MHS member who was diagnosed with Mpox between 4 May 2022 and 10 November 2022. Including retrospective aggregative information about all infected MHS patients, the study was designed to allow for an overall estimation of Mpox in Israel, while considering the results of the detailed patient journeys in a comparable subset of that population. MHS has kept a centralized database of EMRs for over three decades. This database includes longitudinal data of sociodemographic information, anthropometric measurements, outpatient and hospital diagnoses, procedures and consultations, medication prescriptions and purchases, imaging performed, comprehensive laboratory data from a single central laboratory, and locally developed automated registries.

In this study, individual-level data included demographics, namely biological sex, age, social sector, and socioeconomic status (SES). The SES, assigned by Israel’s Central Bureau of Statistics [6], is measured on a scale from 1 (lowest) to 10, and the index is calculated based on several parameters, including household income, educational qualifications, household crowding, and car ownership. Data were also collected on chronic diseases from MHS’ automated registries, including cardiovascular diseases [7], diabetes [8], inflammatory bowel diseases, and cancer, from the National Cancer Registry [9].

This study is descriptive in nature. Statistics are presented as frequencies (n, %) for categorical variables, and as mean (standard deviation SD) and median (interquartile range) for continuous variables.

## 3. Results

### 3.1. Characteristics of All Mpox Patients in MHS

Overall, there were 159 MHS members diagnosed with an Mpox infection between 4 May 2022 and 10 November 2022, mostly between July and September of 2022 (Figure 1). Thus, 90.6% of infected individuals were males, and the mean age at diagnosis was 36.3 years (SD 12.5) (Table 1). Most patients were living in central, non-peripheral areas of Israel (96.9%) and were of high (57.9%) and middle (39.6%) socioeconomic status.

Diagnosis was confirmed through an MPXV PCR test, reported back to MHS’ central database as either a numerical value of cycle threshold (Ct) or a binary value (positive/negative). Thus, 63 out of 159 infected individuals (39.6%) had numerical results from PCR tests, with a mean Ct value for an Mpox-specific target (West African (WA) clade) for any anatomical site at 27.5 (Table 2). Mean Ct values were lower in samples from skin lesions (pustule fluid) (24.9) than for rectal swabs (26.6) and oropharyngeal swabs (32.2) (Figure 2).

As for other comorbidities of infected individuals, 16 out of 159 patients (10.1%) were diagnosed with HIV (Human Immunodeficiency Virus), and all were treated with anti-viral medications. Of those without HIV, 70 (44%) were using pre-exposure prophylaxis (or PrEP). Further, 99 (62.3%) patients with Mpox had a history of sexually transmitted infections (STIs), such as syphilis, chlamydia and condylomata acuminata, and gonorrhea, and 22 individuals had concomitant STIs, mostly gonorrhea and syphilis. Other comorbidities were relatively uncommon, with 6.3% obesity and 8 (5%) patients diagnosed with hypertension. The characteristics of the 18 clinical trial participants who underwent interviews to characterize their symptoms compared to the general MHS population infected with Mpox can be seen in Table 1.

We found 15 females infected with Mpox. When comparing male to female Mpox patients (Table 3), it appears that infected females were older, with a mean age of 46 compared to 35.3 for males. Correspondingly, they also had a somewhat higher rate of general comorbidities. Contrastingly, they suffered less from STIs.

Lastly, in August 2022, Israel started vaccinating against Mpox with the smallpox and Mpox vaccine (JYNNEOS; Modified Vaccinia Ankara-Bavarian Nordic; MVA-BN), with 2348 MHS members vaccinated by 10 November 2022. Further, 32 of those vaccinated MHS members were infected; 20 were infected after vaccination (breakthrough infection); 12 were infected and then vaccinated. Additionally, 65% of the breakthrough infections occurred on days 0–14 after inoculation.

### 3.2. Detailed Patient Journey of 18 Individuals

#### 3.2.1. Exposure and Journey to Diagnosis

All 18 individuals reported having sexual intercourse within two months prior to Mpox infection; 16 of them reported having unprotected sexual intercourse (Table 4). Fifteen individuals agreed to share more information with the researchers. Of them, 14 had unprotected sexual relations with a man in the month leading to Mpox infection, 1 reported unprotected sex with a woman, 11 individuals reported having unprotected oral sexual intercourse prior to diagnosis, and the same number reported unprotected anal sex. When specifically asked about known exposure, 10 individuals reported known exposure to individuals with Mpox, all in the form of sexual intercourse.

Interestingly, in 72% of cases, suspicion of Mpox was first raised by the patients themselves, whereas in only 17% was a physician the first to suspect it. It took a median of 3.5 days from first symptom to first Mpox suspicion. We were able to draw a complete patient journey from exposure to disease in 6 of the 18 individuals, where the median time from exposure to first symptom was 5.5 days (range 3–27), and that from symptom to test was 6.5 days. Results took an additional 2 days to be reported (Figure 3).

#### 3.2.2. Mucocutaneal Manifestations

We observed that 17 out of 18 infected individuals with Mpox had mucocutaneal lesions; 8 of those 17 (47%) had less than 10 lesions, 8 (47%) had 10–100 lesions, and 1 had over 100 lesions (Table 4). About one-third experienced a single lesion as the first mucocutaneal manifestation. Further, 58.8% of lesions were developed asynchronously, while only two patients had lymphadenopathy prior to the appearance of lesions. Most patients (82%) had lesions in two or more different anatomical regions and 35% in five or more locations, with 71% in the head and neck area, 53% with lesions over the chest, abdomen, or back, 71% over the arms or legs, and two patients (12%) over the hands and feet. Nine patients (53%) suffered penile lesions, and six (35%) had anal or perianal lesions. Only one patient reported oropharyngeal lesion. As can be seen in Figure 4, the duration of lesions did not alter significantly by their anatomical location. In terms of lesion evolution, papules appeared early and were the first to disappear with pruritis, followed by vesicles and pustules (Figure 5). Crusts were the last to appear and disappear, continuing after the disappearance of pruritis.

#### 3.2.3. Other Symptoms

All individuals experienced constitutional symptoms, and fatigue was reported by all (Table 4). Furthermore, 83% of Mpox patients had fever, which started early and lasted for up to two weeks (Figure 6). Headache and myalgia were common as well and followed the same pattern. Lymphadenopathy was present in 78% of patients, mostly in the inguinal region. Other less common symptoms included arthralgia and back pain, sore throat, dysuria, rectal pain, and constipation. Though limited by patient numbers, it can be seen that higher CT values led to diseases with fewer symptoms and to a shorter duration of symptomatic disease (Figure 7).

Anxiety related to the disease accompanied 13 of the infected individuals, often to a significant degree (with a median degree of ‘4’ on a scale of 0 to 5), lasting throughout symptom duration. Additionally, Mpox patients revealed during interviews that there is significant anxiety as to the disease course and its reporting among GBMSM, and that many individuals do not report symptoms due to fear of stigma and lack of curative treatment.

## 4. Discussion

This descriptive study characterized Mpox in Israel, while constructing a detailed patient journey from multiple in-depth interviews with infected individuals. Our data showed that most of those infected in Israel were young men; however, the median age of 35 was slightly lower than other reports published from other countries, where it leaned towards the late 30 s [10,11,12] and early 40s [13,14,15]. Though younger than these previously published reports, the minimal age of 27.3 still seems not low enough, which could possibly be explained by underreporting and fear of stigma, relating to information obtained during interviews, alongside the fact that no treatment exists, and a long period of self-quarantine is mandated. An additional explanation could relate to the prevalence of asymptomatic carriers, perhaps in younger populations. The existing literature has varied on this topic, whereas some report asymptomatic infections as contributing to the transmission chain [16], while others underscore it [17], stressing the need for further studies.

Interestingly, it seems that fewer Mpox patients in Israel are people living with HIV (PLHIV) compared to other countries, where percentages ranged from 24% to 35% in London [10,13] to 40–57% in Spain [11,14] and 41% in a multi-country analysis [12]. This could be attributed to historically lower HIV rates in Israel compared to other Western countries [18], though rates in more recent years have increased [19]. Nonetheless, PLHIV who suffered Mpox infection were mostly treated with antiretroviral medications at the time of infection, both in Israel and in other locations. Concomitant STIs were less common in MHS [12,15], whereas past STIs were underreported in other studies, so our relatively high percentage could not be assessed comparatively.

Over half of the participants in our clinical trial were able to identify a sexual partner infected with Mpox, with whom they had contact prior to developing symptoms, a higher proportion of a defined exposure event than identified in previous studies. This might be explained due to the study design by which personal relationship and trust was built between participants and the research team that interviewed them and followed them daily.

Most participants reported unprotected sexual intercourse with a man one and two months prior to becoming infected, which matches other reports around the world [15] and supports further investigations as to the still-unanswered question of whether Mpox should be considered a sexually transmitted disease [20,21].

The median incubation period in Israeli infected patients was 5.5 days, which is, overall, within the range reported around the world, though the latter was wide and varied from 2 to 30 [11,12,13,14,15]. Interestingly, and undocumented in previous studies, most patients raised a suspicion of Mpox themselves, whereas in only 17% of cases, it was the physician who first suspected the disease, which could point to higher percentages than previously reported, such as an online survey performed in the Netherlands, demonstrating that only half of the 394 participants were able to self-diagnose [22]. The median time from symptom to first suspicion was 3.5 days. With that, the median time from first symptom to a confirmatory test was 6.5 days, which when added to the incubation period, could explain the rapid spread of the disease prior to mandatory self-isolation.

Most infected individuals in our study had mucocutaneal lesions, about half with more than 10 lesions, higher than reported by other studies [10,11,12]. Lesions were often found in the anogenital regions, characterizing the disease; however, in our sample, lesions were rather more widespread with lesions over the head and neck, trunk, and limbs than other reports [12], though less common in the oropharyngeal region [11]. Lesions developed asynchronously with varying morphologies, matching the atypical presentation of Mpox previously reported [4,23,24], hypothesized to be attributed to a phenomenon of possible autoinoculation [10]. We found that duration of lesions did not alter significantly by their anatomical location, which we could not find specific consideration of in the existing literature.

Constitutional symptoms were present in all participants, surpassing some previous reports [10,13] but matching others [14,15], where, in general, fever was more common in our sample. Other common symptoms included fatigue, headache, and myalgia, all previously reported.

Mean Ct values were lower in samples from skin lesions (pustule fluid) than for rectal swabs and oropharyngeal swabs, which has been reported in other studies as well [11,12,25]. We also found that lower Ct values seemed to correlate with more symptoms and longer duration, though findings were limited by patient numbers. This correlation was not previously reported. Such findings could be explained either by a correlation between lower Ct values and a more virulent disease or by the timing of sampling relative to disease progression. Larger studies are needed to distinguish between these hypotheses and further verify this new finding.

Anxiety was uncommonly reported in previous publications [26], while in our sample, it was both common as well as reported to a high degree (a median mark of 4/5), lasting throughout symptom duration. The underreporting of this symptom in other studies should be addressed in future research, as called out by healthcare specialists before, and examined whether it diminishes as Mpox becomes less stigmatized and better clinically understood [27,28]. Furthermore, educational efforts about the disease, its symptoms, and typical patient journey should be provided to both healthcare practitioners [29] as well as patients; particularly, those at higher risk such as GBMSM should be carried out on a large scale in a systematic way. With that, efforts to reduce stigma [28], such as renaming the condition and educating the public, are important, both from the viewpoint of mental health concern of the individual [27] as well as from healthcare management perspective, to encourage testing and seeking medical care. Emphasis should also be placed on rapid detection, which was lagging in our findings, with a low threshold for testing, followed by mandatory self-isolation of infected patients in order to stop the spread of this and future outbreaks of infectious diseases, with the importance of social responsibility in this context further stressed.

Our study is subject to limitations, stemming primarily from our sample size, limiting our ability to carry out non-descriptive analyses. Furthermore, systemic screening of the population was not carried out; therefore, we could be underestimating asymptomatic infections or populations that experienced symptoms but chose to avoid medical assistance and were, therefore, not tested and, thus, not included in this analysis.

## 5. Conclusions

In conclusion, our study demonstrated that clinical trials in the format of in-depth interviews and a long-term relationship with the medical researcher contribute greatly to a deeper understanding of patient journey, especially in new or stigmatized diseases. New emerging infections, such as Mpox, should be further investigated to assess asymptomatic carriers, especially when rapid spreading is observed, even if the clinical presentation, when symptomatic, is easily identifiable. Emphasis should also be placed on rapid detection and education for self-quarantine. Further studies should examine whether lower Ct values correlate with the number of Mpox symptoms and a longer duration of them.

## Figures and Tables

**Figure 1 microorganisms-11-01042-f001:**
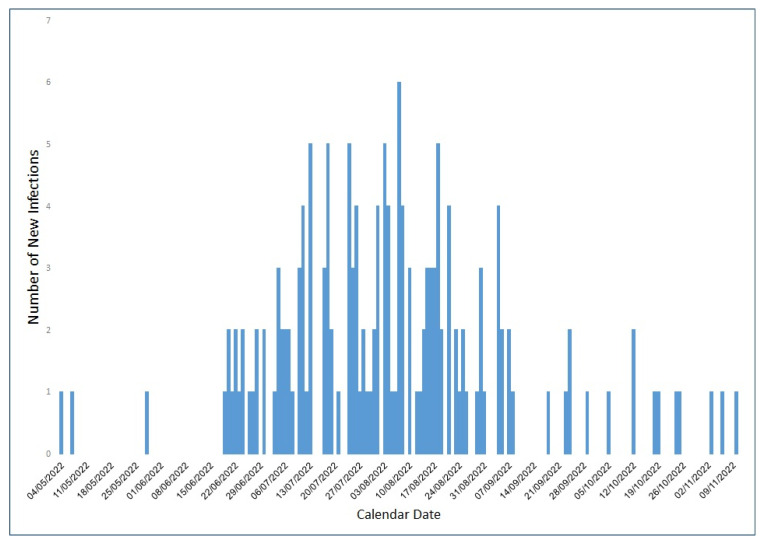
Number of MHS members infected with Mpox between May and November 2022, by dates.

**Figure 2 microorganisms-11-01042-f002:**
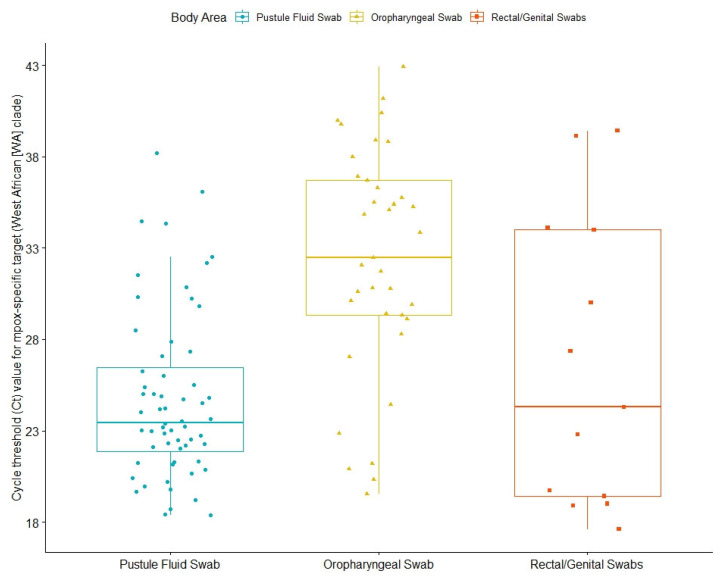
Cycle threshold (Ct) values for Mpox-specific targets (West African (WA) clade) of Mpox patients. The box plot indicates the interquartile range and median value; the whiskers indicate extreme data points.

**Figure 3 microorganisms-11-01042-f003:**
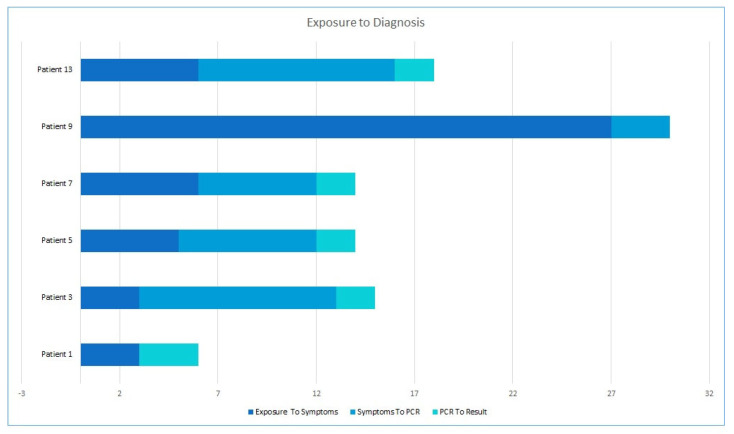
Patient journey from exposure through symptoms and test to results.

**Figure 4 microorganisms-11-01042-f004:**
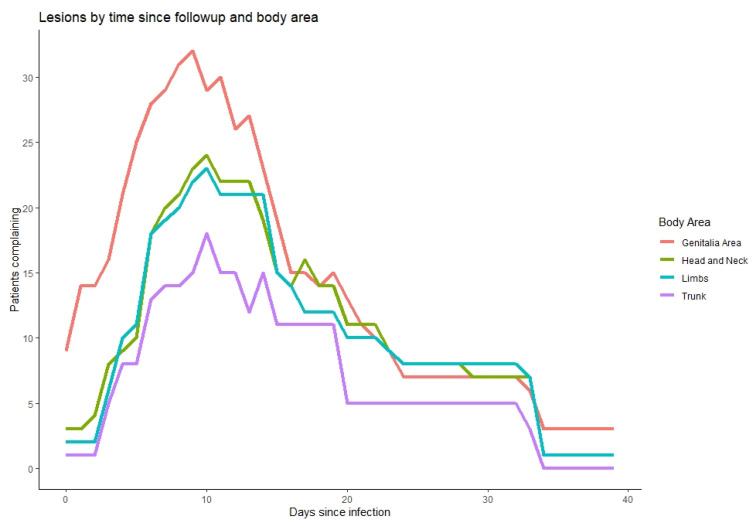
Timing of lesions (x axis) according to anatomical site (colors, legend), relative to the number of patients who reported them on that day (y axis).

**Figure 5 microorganisms-11-01042-f005:**
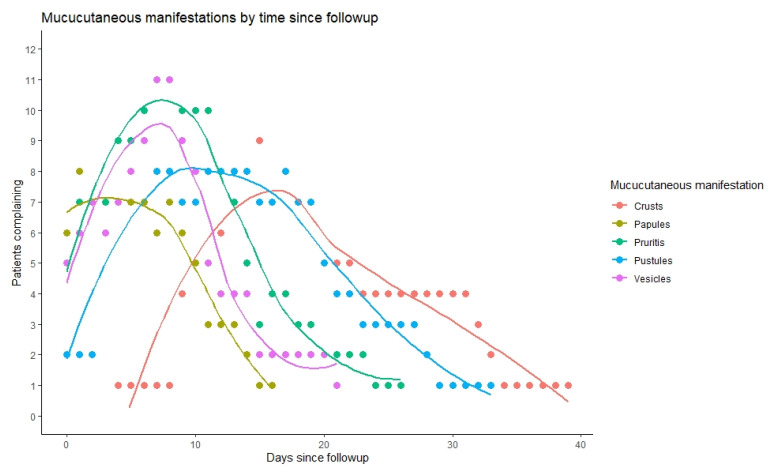
Morphological evolution of mucocutaneus lesions by days since infection (x axis), relative to the number of patients who reported them on that day (y axis).

**Figure 6 microorganisms-11-01042-f006:**
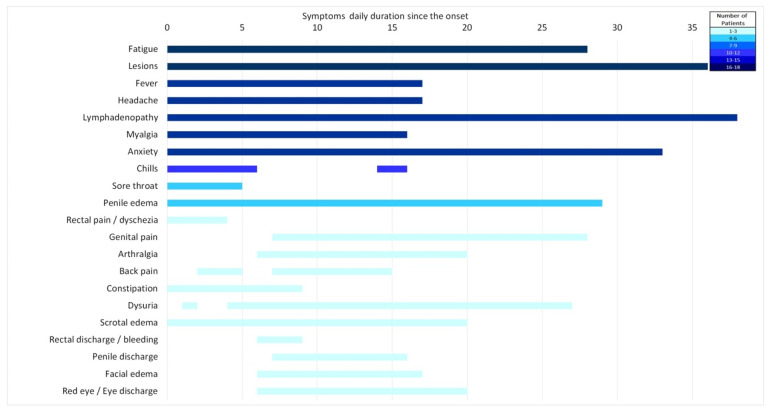
Duration of symptoms, relative to the number of patients who reported them on that day (color coded, legend).

**Figure 7 microorganisms-11-01042-f007:**
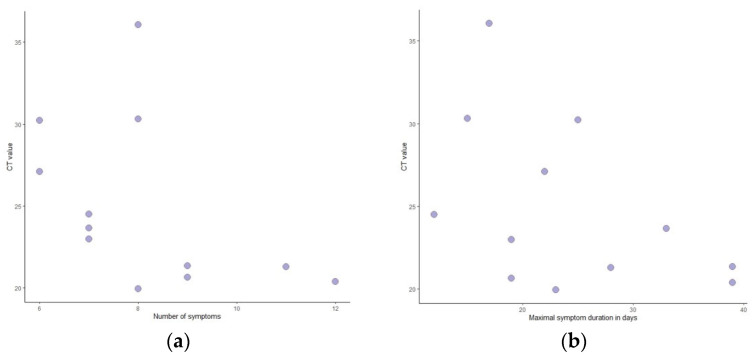
Ct values relative to number of symptoms (**a**) and duration of symptoms (**b**).

**Table 1 microorganisms-11-01042-t001:** Characteristics of individuals infected with Mpox between 4 May 2022 and 10 November 2022.

	Clinical Trial Participants	All MHS Mpox Patients
	(N = 18)	(N = 159)
**Demographics**		
Sex—n (%)		
Female	0 (0%)	15 (9.4%)
Male	18 (100%)	144 (90.6%)
Age at Mpox diagnosis		
Mean (SD)	37.6 (6.27)	36.3 (12.5)
Median [Min, Max]	36.6 [27.3, 58.2]	35.0 [0.970, 84.7]
Socioeconomic status—n (%)		
High (7–10)	13 (72.2%)	92 (57.9%)
Middle (4–6)	5 (27.8%)	63 (39.6%)
Low (1–3)	0 (0%)	4 (2.5%)
Periphery indicator—n (%)		
No	18 (100%)	154 (96.9%)
Yes	0 (0%)	5 (3.1%)
**Comorbidities—n (%)**		
Diabetes Mellitus (Yes)	0 (0%)	1 (0.6%)
Hypertension (Yes)	1 (5.6%)	8 (5.0%)
Cardiovascular diseases (Yes)	0 (0%)	5 (3.1%)
Obesity (BMI ≥30) (Yes)	0 (0%)	10 (6.3%)
**Concomitant Sexually Transmitted Infection (STI)—n (%)**		
Any concomitant STI (Yes)	4 (22.2%)	22 (13.8%)
Chlamydia (Yes)	0 (0%)	2 (1.3%)
Condyloma Acuminatum (Yes)	1 (5.6%)	3 (1.9%)
Gonorrhea (Yes)	2 (11.1%)	7 (4.4%)
Mycoplasma genitalium (Yes)	0 (0%)	1 (0.6%)
Syphilis (Yes)	1 (5.6%)	8 (5.0%)
Herpes Genitalis (Yes)	0 (0%)	1 (0.6%)
**HIV status and antiretroviral treatment**		
HIV positive (Yes)	4 (22.2%)	16 (10.1%)
Antiretroviral treatment among HIV (Yes)	4 (100%)	16 (100%)
Use of preexposure prophylaxis (PrEP) against HIV in HIV negative (Yes)	12 (66.7%)	70 (44.0%)
**Prior Sexually Transmitted Infection—n (%)**		
Any prior STI (Yes)	14 (77.8%)	99 (62.3%)
Chlamydia (Yes)	5 (27.8%)	35 (22.0%)
Condyloma acuminatum (Yes)	7 (38.9%)	41 (25.8%)
Gonorrhea (Yes)	8 (44.4%)	35 (22.0%)
Mycoplasma genitalium (Yes)	2 (11.1%)	15 (9.4%)
Syphilis (Yes)	4 (22.2%)	28 (17.6%)
Herpes genitalis (Yes)	2 (11.1%)	6 (3.8%)
Phthirus pubis (Yes)	5 (27.8%)	15 (9.4%)

**Table 2 microorganisms-11-01042-t002:** MPXV PCR Ct value results of swabs from different anatomical lesions.

	Clinical Trial Participants	All MHS Mpox Patients
	(N = 18)	(N = 159)
**Ct Values**		
Overall mean Ct		
Mean (SD)	28.0 (4.56)	27.5 (4.74)
Median [Min, Max]	27.4 [21.1, 36.1]	27.3 [18.4, 38.8]
Missing	6 (33.3%)	96 (60.4%)
Oropharyngeal swab		
Mean (SD)	31.6 (7.90)	32.2 (6.31)
Median [Min, Max]	33.6 [19.6, 40.4]	32.3 [19.6, 42.9]
Missing	10 (55.6%)	123 (77.4%)
Skin lesion swab		
Mean (SD)	24.9 (5.05)	24.9 (4.69)
Median [Min, Max]	23.3 [19.9, 36.1]	23.5 [18.4, 38.2]
Missing	6 (33.3%)	108 (67.9%)
Rectal swabs		
Mean (SD)	34.1 (NA)	26.6 (7.96)
Median [Min, Max]	34.1 [34.1, 34.1]	24.3 [17.6, 39.4]
Missing	17 (94.4%)	146 (91.8%)

**Table 3 microorganisms-11-01042-t003:** Characteristics of males and females infected with Mpox between 4 May 2022 and 10 November 2022.

	Females	Males	All
	(N = 15)	(N = 144)	(N = 159)
**Demographics**			
Age at Mpox diagnosis			
Mean (SD)	46.0 (23.1)	35.3 (10.5)	36.3 (12.5)
Median [Min, Max]	48.3 [0.970, 84.7]	34.3 [2.84, 72.8]	35.0 [0.970, 84.7]
Socioeconomic status—n (%)			
High (7–10)	7 (46.7%)	85 (59.0%)	92 (57.9%)
Middle (4–6)	8 (53.3%)	55 (38.2%)	63 (39.6%)
Low (1–3)	0 (0%)	4 (2.8%)	4 (2.5%)
Periphery indicator—n (%)			
No	15 (100%)	139 (96.5%)	154 (96.9%)
Yes	0 (0%)	5 (3.5%)	5 (3.1%)
**Comorbidities—n (%)**			
Diabetes Mellitus (Yes)	1 (6.7%)	0 (0%)	1 (0.6%)
Hypertension (Yes)	2 (13.3%)	6 (4.2%)	8 (5.0%)
Cardiovascular diseases (Yes)	2 (13.3%)	3 (2.1%)	5 (3.1%)
Obesity (BMI ≥30) (Yes)	1 (6.7%)	9 (6.2%)	10 (6.3%)
**Concomitant Sexually Transmitted Infection—n (%)**			
Chlamydia (Yes)	0 (0%)	2 (1.4%)	2 (1.3%)
Condyloma Acuminatum (Yes)	0 (0%)	3 (2.1%)	3 (1.9%)
Gonorrhea (Yes)	0 (0%)	7 (4.9%)	7 (4.4%)
Mycoplasma genitalium (Yes)	0 (0%)	1 (0.7%)	1 (0.6%)
Syphilis (Yes)	0 (0%)	8 (5.6%)	8 (5.0%)
Herpes Genitalis (Yes)	0 (0%)	1 (0.7%)	1 (0.6%)
**HIV status and antiretroviral treatment**			
HIV positive (Yes)	0 (0%)	16 (11.1%)	16 (10.1%)
Antiretroviral treatment among HIV (Yes)	0 (100%)	16 (100%)	16 (100%)
Use of preexposure prophylaxis (PrEP) against HIV in HIV negative (Yes)	0 (0%)	70 (48.6%)	70 (44.0%)
**Prior Sexually Transmitted Infection—n (%)**			
Any prior infection (Yes)	4 (26.7%)	95 (66.0%)	99 (62.3%)
Chlamydia (Yes)	0 (0%)	35 (24.3%)	35 (22.0%)
Condyloma acuminatum (Yes)	2 (13.3%)	39 (27.1%)	41 (25.8%)
Gonorrhea (Yes)	0 (0%)	35 (24.3%)	35 (22.0%)
Mycoplasma genitalium (Yes)	0 (0%)	15 (10.4%)	15 (9.4%)
Syphilis (Yes)	0 (0%)	28 (19.4%)	28 (17.6%)
Herpes genitalis (Yes)	1 (6.7%)	5 (3.5%)	6 (3.8%)
Phthirus pubis (Yes)	0 (0%)	15 (10.4%)	15 (9.4%)

**Table 4 microorganisms-11-01042-t004:** Detailed exposure and symptomatology information of 18 individuals with Mpox.

	N	%
**Exposure**		
**Traveling abroad within a month prior to Mpox diagnosis**		
No	6	33%
Yes	6	33%
**Known exposure to Mpox**	10	56%
**Sexual history**		
Sexual intercourse within two months prior to the disease	18	100%
**Sexual history in the month prior to diagnosis**		
Unprotected Sexual intercourse	16	89%
Unprotected sexual intercourse with men	14	78%
Unprotected sexual intercourse with women	1	6%
Unprotected oral sexual intercourse	11	61%
Unprotected anal sexual intercourse	11	61%
**First suspicion raised by**		
The patient	13	72%
A close contact (friend or family)	1	6%
The primary care physician	1	6%
An emergency department physician	2	11%
**Time from symptom to first suspicion (days) median [min, max]**	3.5	[0, 10]
**Symptoms**		
**Mucocutaneal lesions**		
** Number of regions**		
1 region	3	17%
2–5 regions	8	44%
≥5 regions	6	33%
** Lesion location**		
Face	8	44%
Neck	4	22%
Shoulders	1	6%
Chest	2	11%
Abdomen	4	22%
Back	3	17%
Buttocks	0	0%
Arms/legs	12	67%
Hands/Feet	2	11%
Groin	7	39%
Penis	9	50%
Scrotum	0	0%
Anal/perianal	6	33%
Oropharynx	1	6%
** Pain of lesions**	1	6%
**Constitutional symptoms (-any)**	18	100%
Fever	15	83%
Chills	11	61%
Fatigue	18	100%
Headache	14	78%
Myalgia	13	72%
Back pain	2	11%
Arthralgia	2	11%
**Lymphadenopathy (-any)**	14	78%
Submandibular	2	11%
Cervical	4	22%
Axillary	0	0%
Inguinal	14	78%
**Sore throat**	4	22%
**Eye pain, red eye or discharge**	1	6%
**Facial edema**	1	6%
**Fear of the disease [0, 1–5] (median) (min., max.)**	4	[2, 5]
**Rectal pain/dyschezia**	5	28%
**Rectal discharge or bleeding**	1	6%
**Constipation**	2	11%
**Dysuria**	2	11%
**Pain in genitals**	1	6%
**Penile edema**	5	28%
**Penile pruritis**	4	22%
**Penile discharge**	1	6%
**Edema of scrotum**	2	11%

## Data Availability

According to the Israel Ministry of Health regulations, individual-level data cannot be shared openly. Specific requests for remote access to de-identified community-level data or code used should be referred to KSM, Maccabi Healthcare Services Research and Innovation Center.

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
