# Peer review of "Mpox Patient Journey in Israel"

_microorganisms, 2023, doi:10.3390/microorganisms11041042_

Round 1
Reviewer 1 Report
The aim of this study was to characterize Mpox in the Israeli population during the recent (2022) outbreak. A unique aspect of this report is that the authors intentionally investigated patient health histories and contact tracing during disease. One primary weakness of the report is the lack of methods associated with the PCR results. Please see the following comments/recommendations:
Line 13: Individual needs to be pluarlized (i.e., "individuals")
Line 17: The phrase "...and to a a test..." is unclear. Please correct.
Line 27: Suggest to indicate "formerlly referred to as monkeypox" in the parentheses
Line 28: Rather than state "pre-2022", please be more specific and include date range(s)
Line 46: This section requires additional method description(s) for the PCR. Please address.
Line 69: Please define "EMRs" earlier
Lines 73-75: The referenced registries here are relevant, but this is essentially repeated in the following section (Lines 80-82). Please delete Lines 73-75, but keep Lines 80-82.
General note throughout: On numerous instances, sentences are started with a number, value, or percentage. This is not the preferred approach. Please reword these sentences such that the value is embedded rather than starting the sentence. For example, Line 90 can read as follows: "Of the infected individuals, 90.6% were..."
Figure 1: Please add another tick to the y-axis (i.e., tick value should equal 1)
Figure 1: Please clarify if these are new infections per day
Line 99: More details are needed on the MPXV PCR test. Is this a kit? Manufacturer? Testing conditions, etc. Were samples analyzed in a centralized laboratory?
Line 102: West African is misspelled.
Figure 2: Please revise the y-axis to include ticks at the very top and very bottom. In addition, please describe the bars (assuming mean, but they could be median) and error bars. Finally, the y-axis title (Ct-WA) is unclear. Please revise or better define.
Figure 3: Please expand the figure legends (i.e., do not abbreviate).
Figure 5: The Y-axis requires more ticks. Also, see "Crusts"; you cannot have less than 0 infected individuals; please correct. Also, how were the trend lines formed on Figure 5? They almost look hand-drawn; if so, this is unacceptable and a mathematical relation should be added.
Line 185: There are issues with Figure 7 legend (Italicized Figure 7, misspelled "symptoms", etc.)
Line 189: The statement regarding anxiety in the gay community is likely accurate, however it is too broad in context of this manuscript. Please be more specific. Perhaps indicate that there is much anxiety related to disease reporting (particularly STDs) or something similar. Also, you may refrain from specifying "gay" since your manuscript uses the term "GBMSM."
Lines 192-194: The first sentence of the Discussion is unclear. Please reword.
Author Response
Response to reviewer #1, Microorganisms
Ms. no.: microorganisms-2321348 - Mpox patient journey in Israel
Comments and Suggestions for Authors
The aim of this study was to characterize Mpox in the Israeli population during the recent (2022) outbreak. A unique aspect of this report is that the authors intentionally investigated patient health histories and contact tracing during disease. One primary weakness of the report is the lack of methods associated with the PCR results.
We thank the reviewer for their comments and thoughtful consideration of our manuscript. Detailed responses below.
Please see the following comments/recommendations:
- Line 13: Individual needs to be pluarlized (i.e., "individuals")
- Thank you, corrected.
- Line 17: The phrase "...and to a a test..." is unclear. Please correct.
- Line 27: Suggest to indicate "formerlly referred to as monkeypox" in the parentheses
- We accept the reviewer’s suggestion. Please see there.
- Line 28: Rather than state "pre-2022", please be more specific and include date range(s)
- Line 46: This section requires additional method description(s) for the PCR. Please address.
- Line 99: More details are needed on the MPXV PCR test. Is this a kit? Manufacturer? Testing conditions, etc. Were samples analyzed in a centralized laboratory?
- All MPXV PCR tests were performed by a central lab operated through the Israeli Ministry of Health, and not through the different HMOs. The HMOs (the health funds) such as Maccabi Healthcare Services, received the results, as we mentioned in the manuscript: “Diagnosis was confirmed by a MPXV PCR test, reported back to MHS’ central database”. We attempted to contact the Israeli Ministry of Health to inquire as to the reviewer’s question, but no response has been received yet, and we cannot account for when it will be. We should mention that other Israeli studies that have been published in peer-review journals used the same PCR methods, such as this vaccination effectiveness study in Nature Medicine: Wolff Sagy, Y., Zucker, R., Hammerman, A. et al. Real-world effectiveness of a single dose of mpox vaccine in males. Nat Med 29, 748–752 (2023). https://doi.org/10.1038/s41591-023-02229-3
- Line 69: Please define "EMRs" earlier
- Lines 73-75: The referenced registries here are relevant, but this is essentially repeated in the following section (Lines 80-82). Please delete Lines 73-75, but keep Lines 80-82.
- Changed according to the reviewer’s suggestion.
- General note throughout: On numerous instances, sentences are started with a number, value, or percentage. This is not the preferred approach. Please reword these sentences such that the value is embedded rather than starting the sentence. For example, Line 90 can read as follows: "Of the infected individuals, 90.6% were..."
- As relayed to the Editor, if accepted, the manuscript will be sent to English editing by the Microorganisms editorial team.
- Figure 1: Please add another tick to the y-axis (i.e., tick value should equal 1)
- Figure 1: Please clarify if these are new infections per day
- These are new infections, y axis was changed accordingly.
- Line 102: West African is misspelled.
- Thank you, corrected.
- Figure 2: Please revise the y-axis to include ticks at the very top and very bottom. In addition, please describe the bars (assuming mean, but they could be median) and error bars. Finally, the y-axis title (Ct-WA) is unclear. Please revise or better define.
- Figure 3: Please expand the figure legends (i.e., do not abbreviate).
- Figure 5: The Y-axis requires more ticks. Also, see "Crusts"; you cannot have less than 0 infected individuals; please correct. Also, how were the trend lines formed on Figure 5? They almost look hand-drawn; if so, this is unacceptable and a mathematical relation should be added.
- The lines are not hand drawn. The plot is a scatter plot created by points projected onto the YX plane, where the continuous lines are the spline smoothing / interpolation of these points (this also explains the below-zero value(s), which were removed per the reviewer’s suggestion).
- Line 185: There are issues with Figure 7 legend (Italicized Figure 7, misspelled "symptoms", etc.)
- Line 189: The statement regarding anxiety in the gay community is likely accurate, however it is too broad in context of this manuscript. Please be more specific. Perhaps indicate that there is much anxiety related to disease reporting (particularly STDs) or something similar. Also, you may refrain from specifying "gay" since your manuscript uses the term "GBMSM."
- Changed accordingly.
- Lines 192-194: The first sentence of the Discussion is unclear. Please reword.
- 5 mucocutaneal in legend
Reviewer 2 Report
Thank you for your paper “Mpox patient journey in Israel”. See below my comments:
Line 15. Use electronic medical records in the abstract, not the abbreviation.
Line 62 See above
At Line 115 the authors wrote: “clinical trial participants who underwent interviews to characterize their symptoms, were overall similar to the general MHS population infected with Mpox”. This is not really true. Subjects enrolled in clinical trials were only males with high SES and a lower number of comorbidities. It is not clear which is the aim leading to the comparison of the two groups.
In the conclusions the authors wrote that their study demonstrated that “clinical trials in the format of in-depth interviews and a long-term relationship with the medical researcher contribute greatly a deeper understanding of patient journey, especially in new or stigmatized diseases”.
However, in the current form the paper the paper does not provide any new information with respect to what is already known from the literature. It appears to be a descriptive study based on periodic interviews of only 18 patients. In addition the authors were able to draw a complete patient journey from exposure to disease in only 6 of the 18 individuals. The references are scarce.
Author Response
Response to reviewer #2, Microorganisms
Ms. no.: microorganisms-2321348 - Mpox patient journey in Israel
Comments and Suggestions for Authors
Thank you for your paper “Mpox patient journey in Israel”. See below my comments:
We thank the reviewer for their comments and thoughtful consideration of our manuscript. Detailed responses below. We would also like to mention that if accepted, the manuscript will be sent to English editing by the Microorganism journal editorial team.
- Line 15. Use electronic medical records in the abstract, not the abbreviation.
- Changed according to the reviewer’s suggestion.
- Line 62 See above
- At Line 115 the authors wrote: “clinical trial participants who underwent interviews to characterize their symptoms, were overall similar to the general MHS population infected with Mpox”. This is not really true. Subjects enrolled in clinical trials were only males with high SES and a lower number of comorbidities. It is not clear which is the aim leading to the comparison of the two groups.
- The aim of Table 1 (the comparison of the two groups) was to allow the reader full transparency as to the potential differences between the clinical trial consented participants and the general infected population, so when results are shown and inferences drawn concerning the patient journey, readers these could be read critically, in light of the characteristics of the population compared to the entire population. Furthermore, as to the overall characteristics, the median age, periphery/central living indicator and overall comorbidity profile are similar, as well as their Ct values (Table 2), also please notice that overall small sample size in the clinical trial participant group affects percentages significantly. Nonetheless, following the reviewer’s concern, we revised this sentence, please see there.
- In the conclusions the authors wrote that their study demonstrated that “clinical trials in the format of in-depth interviews and a long-term relationship with the medical researcher contribute greatly a deeper understanding of patient journey, especially in new or stigmatized diseases”. However, in the current form the paper the paper does not provide any new information with respect to what is already known from the literature. It appears to be a descriptive study based on periodic interviews of only 18 patients. In addition the authors were able to draw a complete patient journey from exposure to disease in only 6 of the 18 individuals.
- First, and important to clarify, the fact that we were able to draw a “complete” journey from first exposure through the entire course in only 6 – does NOT mean we did not have complete patient journey since all 18 were infected – which we do. “Complete” refers to time from first exposure. All patients had information from first symptom or first suspicion of infection. Second, it is entirely UNTRUE that the paper does not provide new information, as elaborated extensively in the Discussion, but summarized partially here as well:
- As for NEW Israel-specific information: (1) the population in Israel was younger, unlike previously reported; (2) less Mpox patients in Israel are people living with HIV (PLHIV) compared to other countries; (3) Concomitant STIs were less common than in other countries.
- As for NEW information relevant globally: (1) We found that lower Ct values correlated with more symptoms and longer duration of them – a correlation that was not previously reported; (2)We report history of past STIs in infected Mpox patients, which was unpublished; (3) We report that most Mpox patients raised the suspicion of Mpox themselves, whereas in only 17% of cases it was the physician who first suspected the disease – this was also undocumented in previous studies; (4) Anxiety was uncommonly reported in previous publications, while in our sample it was both common as well as reported to a high degree (a median mark of 4/5), lasting throughout symptom duration.
- Given all this, we are unsure what the reviewer is basing their comment on.
- First, and important to clarify, the fact that we were able to draw a “complete” journey from first exposure through the entire course in only 6 – does NOT mean we did not have complete patient journey since all 18 were infected – which we do. “Complete” refers to time from first exposure. All patients had information from first symptom or first suspicion of infection. Second, it is entirely UNTRUE that the paper does not provide new information, as elaborated extensively in the Discussion, but summarized partially here as well:
- The references are scarce.
- We included 29 references. We kindly ask the reviewer to indicate on which specific aspect of the current outbreak more references are needed, as we believe we have reviewed the current Mpox outbreak in a comprehensive manner.

Reviewer 3 Report
The authors are presenting an analysis of mpox patient trajectories in Israel; I would like to share the following comments with them
Introduction: The Intro is rather minimal and the research question is hardly developed. I would strongly suggest to extend the Intro to prepare the reader for the analysis conducted below. There are REF that are missing that could help to pave the way for the analysis. For example Wang et al (2022) investigated self-diagnosis abilities among a similar population – which dovetails with your “first suspicion” analysis (but yielded lower self-diagnosis abilities):
Wang, H., d'Abreu de Paulo, K. J., Gültzow, T., Zimmermann, H. M., & Jonas, K. J. (2022). Monkeypox self-diagnosis abilities, determinants of vaccination and self-isolation intention after diagnosis among MSM, the Netherlands, July 2022. Eurosurveillance, 27(33), 2200603.
Limitations should be extended and discussed in more detail. Were Israelis with an Arab background included, or why not?
Minor points
Sound language editing is needed, the MS is full of typos or not fully corrected sentences.
e.g. line 16 but also l. 18 “differ”? l. 149 reported or confirmed, please clarify
Fig. 5 mucocutaneal in legend
Please explain abbreviations at first use, e.g. Ct, EMR, MHS
Figures are not legible – please improve graphics and color scheme, and use larger fonts

Author Response
Response to reviewer #3, Microorganisms
Ms. no.: microorganisms-2321348 - Mpox patient journey in Israel
Comments and Suggestions for Authors
The authors are presenting an analysis of mpox patient trajectories in Israel; I would like to share the following comments with them.
We thank the reviewer for their comments and thoughtful consideration of our manuscript. Detailed responses below.
- Introduction: The Intro is rather minimal and the research question is hardly developed. I would strongly suggest to extend the Intro to prepare the reader for the analysis conducted below. There are REF that are missing that could help to pave the way for the analysis. For example Wang et al (2022) investigated self-diagnosis abilities among a similar population – which dovetails with your “first suspicion” analysis (but yielded lower self-diagnosis abilities): Wang, H., d'Abreu de Paulo, K. J., Gültzow, T., Zimmermann, H. M., & Jonas, K. J. (2022). Monkeypox self-diagnosis abilities, determinants of vaccination and self-isolation intention after diagnosis among MSM, the Netherlands, July 2022. Eurosurveillance, 27(33), 2200603.
- As per the reviewer’s suggestion, we added the reference by Wang, H et al, which admittedly has a shorter Introduction that the one provided in our manuscript. It is important to note that we included many references to existing literature throughout the Discussion section, rather than in the Intro section, in order to follow Microorganism’s information for author’s section, where we were instructed to keep the Introduction section brief. The study is descriptive in nature, and as stated in the Introduction, “reports on Mpox have thus far characterized the disease, but mostly through a single time-point view, rather than a patient journey, except for case reports. The aim of this study is to characterize Mpox in the Israeli setting in general, alongside compiling a detailed patient journey from multiple in-depth interviews of infected individual.” Existing literature that follows such a scheme of detailing patient journey is missing, so we cannot provide much context there. Notwithstanding this, in the Discussion section, we address each specific finding in the context of existing literature (e.g., asynchronous lesions).
- Limitations should be extended and discussed in more detail. Were Israelis with an Arab background included, or why not?
- Yes, they were included. ALL Israelis who are members of the HMO (MHS) were included in the study. We are unsure why the reviewer differentiates this population in their comment. Nonetheless, to attempt to follow the reviewer’s logic, it is possible that certain sectors, such as Arabs or ultra-orthodox Jews were infected but chose not to see a physician and therefore were not discovered, possible due to cultural tensions and fear of stigma. To further stress this, we added this to the limitations Apart from that, we discuss stigma in detail throughout the Discussion and Conclusion sections.
Minor points
- Sound language editing is needed, the MS is full of typos or not fully corrected sentences. e.g. line 16 but also l. 18 “differ”? l. 149 reported or confirmed, please clarify
- As relayed to the Editor, if accepted, the manuscript will be sent to English editing by the Microorganisms editorial team.
- 5 mucocutaneal in legend
- Corrected
- Please explain abbreviations at first use, e.g. Ct, EMR, MHS
- Thank you, corrected through the main text, though not in the abstract. If the editor so chooses, we shall explain abbreviations in the abstract as well.
- Figures are not legible – please improve graphics and color scheme, and use larger fonts
- Following the reviewer’s suggestion, new figures were produced, using larger fonts and improved color scheme.

Round 2
Reviewer 2 Report
I thank the authors for having rapidly responded to my criticisms. I have some objections. The median age of the study is approximately 36 years. This age is not younger than reported worldwide (see ECDC/WHO website). Moreover, a recent study conducted in Israel reported that median age was 35 years with a low proportion of HIV positive subjects (12%) that was similar to your study. (Sheffer R, Savion M, Nuss N, Amitai Z, Salama M. Monkeypox outbreak in the Tel Aviv District, Israel, 2022. Int J Infect Dis. 2023 Mar ;128:88-90. doi: 10.1016/j.ijid.2022.12.023. Epub 2022 Dec 22. PMID: 36566775; PMCID: PMC9773787). Regarding the correlation between Ct-values and severity, I think it might be an interesting information but it should be less emphasized considering the very low number of patients analysed. The authors have declared that “We report history of past STIs in infected Mpox patients, which was unpublished”. This is entirely untrue (for ex. Raccagni AR, Candela C, Mileto D, Canetti D, Bruzzesi E, Rizzo A, Castagna A, Nozza S. Monkeypox infection among men who have sex with men: PCR testing on seminal fluids. J Infect. 2022 Nov;85(5):573-607. doi: 10.1016/j.jinf.2022.07.022). The authors have already recently published an article ( DOI: 10.3390/tropicalmed8010015 ) that depicts two detailed patient journeys of Israeli men in their 30s who were diagnosed in recent months, depicting their symptoms, presumed exposure, and outcomes". I wonder why it was not mentioned given the NEW insights that it could provide.
Author Response
Response to reviewer #2, Round 2 of Revisions, Microorganisms
Ms. no.: microorganisms-2321348 - Mpox patient journey in Israel
Comments and Suggestions for Authors
I thank the authors for having rapidly responded to my criticisms. I have some objections.
We thank the reviewer again for their comments and time. Please see responses below.
- The median age of the study is approximately 36 years. This age is not younger than reported worldwide (see ECDC/WHO website). Moreover, a recent study conducted in Israel reported that median age was 35 years with a low proportion of HIV positive subjects (12%) that was similar to your study. (Sheffer R, Savion M, Nuss N, Amitai Z, Salama M. Monkeypox outbreak in the Tel Aviv District, Israel, 2022. Int J Infect Dis. 2023 Mar ;128:88-90. doi: 10.1016/j.ijid.2022.12.023. Epub 2022 Dec 22. PMID: 36566775; PMCID: PMC9773787).
- Our statement of the age in Israel being “slightly younger” (manuscript, Discussion section) than reported worldwide, was based on published literature to which we referenced, please see specifically:
- Reference #10: Median age was 38
- Reference #11: Median age was 37
- Reference #12: Median age was 38
- Reference #13: Median age was 41
- Reference #14: Median age was 42
- Reference #15: Median age was 40
- As the reviewer did not mention a specific source/doi, we cannot comment on it. Nonetheless, we included a more conservative language. Please see there.
- Furthermore, the fact a different study in Israel has shown the same median age, does not contradict our statement at all; we are referring to a younger age In Israel compared to other countries – as stated in the manuscript: “was slightly lower than that reported in other countries”. If relevant, this reference only reinforces the representativeness of our sample in the context of the Israeli population. The same goes for the proportion of the HIV positive individuals in our sample.
- Our statement of the age in Israel being “slightly younger” (manuscript, Discussion section) than reported worldwide, was based on published literature to which we referenced, please see specifically:
- Regarding the correlation between Ct-values and severity, I think it might be an interesting information but it should be less emphasized considering the very low number of patients analysed.
- We appreciate the reviewer’s opinion on the limitation of this finding given our sample size, which is why in that very paragraph, immediately following the sentence that discuss Ct values - and not solely in the limitation section - we mentioned: “Larger studies are needed to distinguish between these hypotheses, and further verify this new finding.” If the Editor so chooses to add more reservations to our finding, please instruct us further.
- The authors have declared that “We report history of past STIs in infected Mpox patients, which was unpublished”. This is entirely untrue (for ex. Raccagni AR, Candela C, Mileto D, Canetti D, Bruzzesi E, Rizzo A, Castagna A, Nozza S. Monkeypox infection among men who have sex with men: PCR testing on seminal fluids. J Infect. 2022 Nov;85(5):573-607. doi: 10.1016/j.jinf.2022.07.022).
- We thank the reviewer for this reference. Unlike multiple existing reports on concurrent STIs in Mpox patients, most of the published studies about Mpox (including all those to which we referenced) in fact do not include information on past STIs, but only on concurrent ones. The article the reviewer is referring to only mentions “previous STIs” as a row in Table 2, without specification as to which STIs, a specification that is given in our manuscript. Our wording in the Discussion section of the manuscript is: “whereas past STIs were underreported in other studies,” which stands true – it is underreported.
- The authors have already recently published an article ( DOI: 10.3390/tropicalmed8010015 ) that depicts two detailed patient journeys of Israeli men in their 30s who were diagnosed in recent months, depicting their symptoms, presumed exposure, and outcomes". I wonder why it was not mentioned given the NEW insights that it could provide.
- We were not adamant to reference our own previous case report. We are unsure why this is mentioned but will of course yield to the Editor’s decision if they wish this to be included in the literature review.

Reviewer 3 Report
I have read a previous version of the MS and I am happy to see how the authors have improved the MS. I would like to share the following comments with the authors.
I think there was a misunderstanding regarding the Introduction. I did not suggest to cut the current Introduction to the format of a Eurosurveillance paper. The reference to the Wang et al paper was just meant for the self-diagnosis aspect and the authors have included that in the Discussion now - great! Eurosurveillance papers have a different structure and it does not apply here. Thus I would suggest to re-recreate a proper Introduction here. Such an Introduction can still be flashed out and to the point. The current research question should be developed better and should flow naturally from the argument.
The authors provide relevant information about the inclusion criteria (all Israelis registered with the health authorities) in the letter to the editors, but the information is still missing in the paper. I would suggest to include it in the paper as well, as well as explaining the Israeli healthcare landscape a bit better - as many readers may not be familiar with it.
Author Response
Response to reviewer #3, Round 2 of revisions, Microorganisms
Ms. no.: microorganisms-2321348 - Mpox patient journey in Israel
Comments and Suggestions for Authors
I have read a previous version of the MS and I am happy to see how the authors have improved the MS. I would like to share the following comments with the authors.
We thank the reviewer again for their comments, time, and careful consideration. Please see responses below.
- I think there was a misunderstanding regarding the Introduction. I did not suggest to cut the current Introduction to the format of a Eurosurveillance paper. The reference to the Wang et al paper was just meant for the self-diagnosis aspect and the authors have included that in the Discussion now - great! Eurosurveillance papers have a different structure and it does not apply here. Thus I would suggest to re-recreate a proper Introduction here. Such an Introduction can still be flashed out and to the point. The current research question should be developed better and should flow naturally from the argument.
- We thank the reviewer for acknowledging the addition of the Wang paper, and the different structure of the Introduction mandated by different journals. However, we still do not understand what should further be flashed out and flow more naturally, in the stated research question:
- “reports on Mpox have thus far characterized the disease, but mostly through a single time-point view, rather than a patient journey, except for case reports.
- To this point, the aim of this study is to characterize Mpox in the Israeli setting in general, alongside compiling a detailed patient journey from multiple in-depth interviews of infected individual.”
- As we mentioned, existing literature that details Mpox patient journey is still missing, so we cannot provide specific arguments here, nor to apply in retrospect specific findings from our studies and draft them as a specific research questions. The aim of the study was to depict a detailed patient journey, which is missing from existing literature. Nonetheless, in the Discussion section, after compiling and presenting our findings, we address each specific one in the context of existing literature.
- If the reviewer would still like revisions, we kindly ask to be more specific, as well as kindly ask the Editor’s recommendation here, to which we will adhere.
- We thank the reviewer for acknowledging the addition of the Wang paper, and the different structure of the Introduction mandated by different journals. However, we still do not understand what should further be flashed out and flow more naturally, in the stated research question:
- The authors provide relevant information about the inclusion criteria (all Israelis registered with the health authorities) in the letter to the editors, but the information is still missing in the paper. I would suggest to include it in the paper as well, as well as explaining the Israeli healthcare landscape a bit better - as many readers may not be familiar with it.
- We thank the reviewer for this comment, and have thus accordingly included an explanation of the Israeli healthcare landscape in the first paragraph of the Material and Methods.
